# Evaluation of Ecological Service Function of *Liquidambar formosana* Plantations

**DOI:** 10.3390/ijerph192215317

**Published:** 2022-11-19

**Authors:** Jingdong Wu, Mingxu Wang, Tingting Wang, Xinxi Fu

**Affiliations:** 1College of Forestry, Central South University of Forestry and Technology, Changsha 410004, China; 2The Forestry Department of Hunan Province, Changsha 410004, China; 3College of Environmental Science and Engineering, Central South University of Forestry and Technology, Changsha 410004, China

**Keywords:** *Liquidambar formosana*, urban forest, ecosystem services, value evaluation

## Abstract

A *Liquidambar formosana* plantation is a kind of fast-grown forest in the subtropical region, providing a variety of ecosystem services such as superior wood, carbon fixation and oxygen release, and biodiversity maintenance. However, the ecological service function value of *Liquidambar formosana* plantations is not clear. To gain insights into the characteristics and importance of its ecological and economic benefits, the *Liquidambar formosana* plantation in the Tianjiling Forest Farm of Changsha City was taken as the specific research object in this paper. The ecological service function evaluation index system for *Liquidambar formosana* plantations was established based on the relevant research worldwide and the actual situation. The market value method, shadow engineering method, carbon tax method, and other environmental economics methods were used to estimate the value of seven ecological service functions (including organic matter production, carbon fixation and oxygen release, water conservation, soil conservation, soil improvement, air purification, and biodiversity maintenance) of the forest of *Liquidambar formosana*. The results indicated that the total economic value of ecological service function provided by the *Liquidambar formosana* plantation of Changsha was 103,277.82 RMB/(hm^2^·a), and the indirect economic value was 8.47 times that of the direct economic value. Among the seven ecological service functions, the value of carbon fixation and oxygen release was the highest (36,703.33 RMB·hm^−2^·a^−1^), thus suggesting that the *Liquidambar formosana* plantation had strong photosynthesis and significant carbon fixation. This study directly reflects the value of forest ecological service function in the form of currency, which is beneficial to provide more insights into forest ecological service function so as to provide basic data and a scientific basis for the protection, construction, and promotion of the sustainable utilization and development of urban forest resources.

## 1. Introduction

The forest ecosystem is the main body of the terrestrial ecosystem and takes on a critical significance in maintaining and regulating the balance of the terrestrial ecosystem [1,2,3,4]. The forest ecosystem is capable of providing human society with a variety of material products. More importantly, it can provide good ecological environment services for human society. In addition, the forest ecosystem plays a pivotal role in maintaining global climate stability, ecological balance, and economic and social development [5,6]. Thus, the forest ecosystem lays the basis of human survival and development.

Among a variety of services provided by the forest ecosystem for a long time, only a few service functions, including biological resources (e.g., forest products, food, and medicinal materials) and tourism resources, can directly enter the commodity market. However, other ecological service functions (e.g., forest production of organic matters, water conservation, soil conservation, soil improvement, carbon fixation and oxygen release, air purification, and biodiversity maintenance) are difficult to be directly presented in the form of currency in the market [7,8]. People’s insufficient understanding of the service function and importance of the ecosystem is likely to lead to unreasonable development and utilization of forest resources. [9].

In fact, the value of the material products provided by the forest for human beings accounts for only a small part of the service value provided by the forest ecosystem, and the contribution and value of its ecological function for human survival and development is significantly greater than this part [10,11]. Cui et al. (2019) evaluated the ecological service functions of major vegetation types in 11 cities in Shanxi Province, China. The results showed that the forest ecosystem has great ecological value, among which the indirect value accounts for 78.98% of the total value [12]. Zou et al. (2022) evaluated the ecological service value in the ecological public-welfare forest in Jiangxi Province, China. The results showed that ecological public-welfare forest plays an important role in water conservation, soil and water conservation, carbon fixation and oxygen release, and maintenance of biodiversity [13]. Ajad and Rashila’s research also suggested that community forests in Dhading district of Nepal not only bring fresh water, fuelwood, and other resources to local villagers but also have more prominent ecological service functions in water conservation, air purification, soil improvement, and carbon fixation and oxygen release [14].

The urban forest is a forest type closely related to cities. In 1962, the term urban forest was first proposed in a survey of outdoor recreation resources conducted by the Kennedy Administration. In 1965, Professor Erik Jorgensen of the University of Toronto, Canada, put forward the formal concept of the urban forest [15]. With the development of modern industry and the change of energy structure and usage mode, the urban environment is deteriorating day by day. The urban forest plays an important role in maintaining and improving the urban environment and is attracting more and more attention [14,16]. The urban forest not only plays an extremely important role in improving the urban microclimate, maintaining the balance between carbon dioxide and oxygen [17], alleviating the urban “heat island effect” [18], and purifying the air [19,20]. In addition, it can enrich urban characteristics and the urban landscape and provide recreation and social communication space for residents [21]. Compared with other forest types, urban forest is more affected by human factors, and most of them are artificial forests.

*Liquidambar formosana*, i.e., maple, triangle maple, or sweet maple, is an excellent fast-growing and high-yield deciduous broad-leaved tree in subtropical areas. It significantly contributes to windbreak and purification of air pollution for its excellent ability to adapt to the natural environment, rapid growth, and high ornamental value. *Liquidambar formosana* is one of the preferred tree species for tree structure adjustment in plantations, and it is often selected as an urban forest species. In addition, *Liquidambar formosana* has high medicinal value. Its leaves contain a variety of trace elements and volatile oil which can stop bleeding, kill bacteria, and diminish inflammation. Many counties in Fujian Province adopted the afforestation model of Chinese fir mixed with *Liquidambar formosana*, thus increasing the survival rate and preservation rate of afforestation and improving the soil structure [22]. *Liquidambar formosana* plantations have shown good ecological benefits, but their ecological value is still unknown. Costanza et al. (1997) analyzed 16 major types of global ecosystems and proposed 17 ecosystem service functions, such as forest air regulation, hydrological process regulation, soil conservation, and recreation provision [23]. Zhao et al. (2004) divided the service functions of China’s forest ecosystem into four categories: product provision, regulatory function, cultural function, and life support function, and established 13 functional index systems [24]. However, the value evaluation is limited by natural, economic, and social factors and there is no unified evaluation index system and evaluation method in the world at present. Therefore, it is still worth further research to evaluate the ecological service function and economic value of *Liquidambar formosana* plantations objectively and accurately.

To gain insights into the ecological service value of the *Liquidambar formosana* plantation, the *Liquidambar formosana* plantation in Tianjiling Forest Farm of Changsha City was taken as the specific research object. The ecological service function evaluation index system of *Liquidambar formosana* plantations was established based on the relevant research worldwide and the actual situation of the *Liquidambar formosana* plantation in Changsha. The market value method, shadow engineering method, carbon tax method, and other environmental economics methods were used to estimate the value of seven ecological service functions (including organic matter production, carbon fixation and oxygen release, water conservation, soil conservation, soil improvement, air purification, and biodiversity maintenance) of the forest of *Liquidambar formosana*. The results indicated that the total economic value of ecological service function provided by the *Liquidambar formosana* plantation of Changsha was 103,277.82 RMB/(hm^2^·a), and the indirect economic value was 8.47 times that of the direct economic value. Among the seven ecological service functions, the value of carbon fixation and oxygen release was the highest (36,703.33 RMB·hm^−2^·a^−1^). This study directly reflects the value of forest ecological service function in the form of currency, which is conducive to deepening people’s understanding of forest ecological service function so as to lay a basic data and scientific basis for the protection, construction, and promotion of the sustainable utilization and development of urban forest resources.

## 2. Materials and Methods

### 2.1. Study Area

#### 2.1.1. Overview of Natural Geography

The study plot is Tianjiling Forest Farm, located in the experimental forest farm of Nanjiao Academy of Forestry Sciences, Changsha City, Hunan Province (Figure 1). The geographic coordinates are 29°06′40′′ north latitude and 113°01′30′′ east longitude. Tianjiling Forest Farm, the study plot, is a low mountain landform with an altitude of 50–108 m and a relatively gentle terrain with a slope below 20° [25]. The forest farm is a subtropical monsoon humid climate zone, with an annual average temperature of 17.2 °C, an extreme maximum temperature of 40.6 °C and a minimum temperature of −9.5 °C, an annual frost-free period of 272 days, an annual average precipitation of 1411.4 mm, an annual average relative humidity of 80%, and an annual sunshine duration of 1717.34 h (data from Changsha Meteorological Bureau). The soil in the study plot is mainly acidic red soil with Quaternary reticulated parent material, with pH between 4.5 and 5.5, and moderate gravel content (data from Hunan Academy of Geological Research).

#### 2.1.2. Survey of the Study Sample Plot

In this experiment, the two plots of the *Liquidambar formosana* plantation are both 24 years old. The area of the two plots is 20 m × 30 m, and they are numbered as No. I and No. II. Stand survey factors include stand age, canopy density, tree height, density, diameter at breast height, biodiversity, and other indicators. The main indicators are listed in Table 1.

### 2.2. Value Evaluation Index and Method

In accordance with the specific situation of the ecosystem of the *Liquidambar formosana* plantation in Changsha and considering the operability of evaluation, the specific evaluation indexes and methods are determined as listed in Table 2.

### 2.3. Equation of Value Calculation

#### 2.3.1. Organic Matter Production

The market value method is used to estimate its value, which is expressed in Equation (1).
*V* = *B* × *A* × *C*(1)
where *V* is the value of organic matter produced by forest (RMB·a^−1^); *A* is forest area (hm^2^); *B* is net forest growth per unit area [m^3^·hm^−2^·a^−1^]; *C* is the market price of living wood (RMB·m^−3^).

#### 2.3.2. Water Conservation

The categorical statistics method is adopted for calculation, which is expressed as follows:*W* = *T* × *A* × *P*(2)
where *W* is the value of the water source conserved by forest (RMB·a^−1^); *T* is forest water content per unit area (m^3^); *P* is the reservoir project cost (RMB·m^−3^).

#### 2.3.3. Soil Conservation

It is evaluated from three aspects: reducing land abandonment, maintaining soil fertility, and mitigating the disaster of sediment accumulation.

The shadow engineering method is used to calculate the economic value of reducing soil waste, which is expressed in Equation (3).
*U*_soil_ = *AC* (*X*_2_ − *X*_1_)/*ρ*(3)
where *U*_soil_ is the value of forest to reduce land abandonment (RMB·a^−1^); *X*_1_ is the soil erosion modulus of forest land (t·hm^−2^·a^−1^). *X*_2_ is the soil erosion modulus of non-forest land (t·hm^−2^·a^−1^). *C* represent the cost of excavating and transporting earthwork per unit volume of soil (RMB·m^−3^); *ρ* is soil bulk density (t·m^−3^).

According to the market price of fertilizer, the economic benefit of maintaining soil fertility is estimated, which is expressed in Equation (4).
*U* = *A*(*X*_2_ − *X*_1_)(*NC*_1_/*R*_1_ + *PC*_1_/*R*_2_ + *KC*_2_/*R*_3_ + *MC*_3_)(4)
where *N, P, K, M* are soil contents of nitrogen, phosphorus, potassium, and organic matter (%), respectively; *C*_1_, *C*_2_, *C*_3_ are, respectively, the price of diammonium phosphate fertilizer, potassium chloride fertilizer and organic matter (RMB·t^−1^); *R*_1_, *R*_2_, *R*_3_ are nitrogen content of diammonium phosphate fertilizer, phosphorus content of diammonium phosphate fertilizer, potassium content of potassium chloride fertilizer (%), respectively.

The estimation of sediment accumulation mitigation value [26] is shown in Equation (5).
*U*_sediment_ = 24%*A*(*X*_2_ − *X*_1_) × *P*/*ρ*(5)
where *U*_sediment_ is the economic value of forest to mitigate sediment accumulation (RMB·a^−1^); others are consistent with the above.

#### 2.3.4. Air Purification

The shadow engineering method is used to evaluate the value of SO_2_ purification, which is expressed in Equation (6).
*V*_s_ = *S_l_* × *A* × *S*_2_(6)
where *Vs* is the value of absorbed SO_2_ (RMB·a^−1^); *S_l_* is the average SO_2_ uptake per unit area of forest [t·hm^−2^·a^−1^]; *S*_2_ is the cost of reducing SO_2_ in pollution prevention and control projects (RMB·t^−1^); others are consistent with the above.

The method for evaluating the economic value of forest dust catching is expressed in Equation (7).
*V*_f_ = *Z*_1_ × *A* × *Z*_2_(7)
where *V*_f_ is the value of forest dust catching (RMB·a^−1^); *Z*_1_ is the average dust catching capacity of forest per unit area [t·hm^−2^·a^−1^]; *Z*_2_ is the cost of dust catching in pollution prevention and control projects (RMB·t^−1^); others are consistent with the above.

#### 2.3.5. Carbon Fixation and Oxygen Release

The economic value of the carbon fixation and oxygen release function is expressed in Equations (8) and (9) [27].
*U*_carbon_ = *AC*_carbon_ (1.63*RB* + *F*)(8)
where *U*_carbon_ is the economic value of forest carbon fixation (RMB·a^−1^); *C*_carbon_ is the price of carbon fixation (RMB·t^−1^); *R* is carbon content in carbon dioxide (27.27%); *F* is the annual carbon fixation of forest soil per unit area (t), which is determined by actual samples; others are consistent with the above.
*U*_oxygen_ = 1.19*C*_oxygen_ *AB*(9)
where *U*_oxygen_ is the economic value of oxygen released by the forest (RMB·a^−1^); *C*_oxygen_ is the cost of industrial oxygen production or medical oxygen price (RMB·t^−1^); others are consistent with the above.

#### 2.3.6. Soil Improvement

The economic value of the soil improvement function is expressed in Equation (10).
*U*_improvement_ = *J*(*NC*_1_/*R*_1_ + *PC*_1_/*R*_2_ + *KC*_2_/*R*_3_)(10)
where *U*_improvement_ is the economic value of soil improvement by forest litter (RMB·a^−1^); *J* is the annual litter amount of forest (t); *N*, *P*, and *K* are the contents of nitrogen, phosphorus, and potassium in litter (%). *C*_1_ and *C*_2_ are market prices of diammonium phosphate fertilizer and potassium chloride fertilizer respectively (RMB·t^−1^); *R*_1_, *R*_2_, *R*_3_ are nitrogen content of diammonium phosphate fertilizer, phosphorus content of diammonium phosphate fertilizer, potassium content of potassium chloride fertilizer (%), respectively.

#### 2.3.7. Maintenance of Biodiversity

The economic value of the function of biodiversity maintenance is expressed in Equation (11).
*V*_w_ = *S* × *A*(11)
where *V*_w_ is the annual value of biodiversity protected by the stand (RMB·a^−1^); *S* is the value of biodiversity protected per unit area [RMB·hm^−2^·a^−1^]; *A* is the stand area (hm^2^).

### 2.4. Data Processing and Analysis Methods

Excel (office2019, Microsoft, Redmond, WA, USA) and SPSS (SPSS 19.0, IBM, Armonk, NY, USA) were used for data processing and statistical analysis. Excel was used to process basic data and determine the weight of ecosystem services.

## 3. Results

### 3.1. Value Evaluation of Organic Matter in Production

The wood production benefit is one of the direct benefits of forests, and its economic value is usually reflected by the market price of wood. Forest vegetation uses solar energy to synthesize inorganic compounds into organic matter through photosynthesis, which is the most critical function of forest ecosystems. This function provides essential organic matters and products for humans and other organisms [28,29]. The stock volume of living standing trees in Changsha’s *Liquidambar formosana* plantation was estimated based on the collected data of each wood inspection ruler from 2019 to 2021, as listed in Table 3. The average annual net stock of *Liquidambar formosana* forest in Changsha was 12.89 m^3^·hm^−2^·a^−1^, significantly higher than the average annual net stock of local forest (1.1 m^3^·hm^−2^·a^−1^) [30], which also fully indicated that *Liquidambar formosana* forest was a fast-growing and high-yield forest with large annual production. On the basis of the market survey and the standard of local wood price of 850 RMB·m^−3^, Equation (1), was adopted to calculate the value of organic matter produced by the *Liquidambar formosana* plantation as 10,908.33 RMB·hm^−2^·a^−1^.

### 3.2. Value Evaluation of Water Conservation

Considering the particularity and technical operability of the ecosystem of *Liquidambar formosana* plantation in Changsha, the water conservation function and value of *Liquidambar formosana* plantation were primarily indicated in this study from three aspects, including canopy interception, litter, and soil water retention.
(1)Calculation of canopy interception. The *Liquidambar formosana* plantation in Changsha has a large leaf area, and the canopy density is high. To study its canopy interception, the precipitation outside the forest should be analyzed first, especially the distribution characteristics of annual precipitation periods (Figure 2). As can be seen from Figure 2, the distribution of precipitation outside the forest shows great disequilibrium in time and has obvious seasonal characteristics. Precipitation was mainly concentrated in summer and autumn, while winter and spring were dry and rainy. From 2019 to 2021, the total annual precipitation outside the forest was nearly 1395.2 mm. After on-site measurement and data collation, the situation of canopy interception and redistribution of *Liquidambar formosana* forest is illustrated in Figure 3. Although rain water penetration accounts for the highest proportion (71.04%) in the reallocation of *Liquidambar formosana* plantation, the canopy interception of *Liquidambar formosana* plantation is also very significant, accounting for almost one third of the total precipitation. Compared with the average canopy interception rate of subtropical deciduous broad-leaved forest (14.31%) [31], the canopy interception effect of the *Liquidambar formosana* plantation in Changsha is stronger and more conducive to reducing the erosion effect of rainwater on soil.(2)Calculation of litter water holding capacity. The water holding capacity of forest litter is closely related to its storage. According to Zhang Xi et al.’s (2018) statistical data on litter reserves of the *Liquidambar formosana* plantation [21] (Table 4), the total dry weight of litter layer reserves of the *Liquidambar formosana* plantation is 4028.215 kg·hm^−2^. The water absorption capacity of the litter on the forest land surface can be measured on the spot. Combined with the known total dry weight of the litter layer reserves, the water holding capacity of the litter can be calculated (Table 5). The litter layer plays a very important role in the water conservation of forests, which can trap atmospheric precipitation and block surface runoff and scour. At the same time, the decomposition of litter forms soil humus, which can significantly improve soil structure and soil permeability.(3)Calculation of soil water holding capacity. Soil water holding capacity directly affects soil water erosion resistance, and it is an important index reflecting soil ecological function. The forest soil water holding capacity of the *Liquidambar formosana* plantation is listed in Table 6. The water storage capacity of forest soil is closely related to soil thickness and soil porosity. To be specific, soil non-capillary porosity is the main channel of soil gravity water movement, which is closely related to soil water storage capacity. Therefore, in this study, the soil water holding state is calculated according to the effective water storage, and the average value is 319.18 t/hm^2^.

Based on the above, the shadow price of reservoir storage capacity is used to calculate the value of forest water conservation, and the cost per unit storage capacity is 5.714 RMB/m^3^ [32], so the annual economic value of water conservation of *Liquidambar formosana* plantation is about 24,415 RMB/(hm^2^·a).

### 3.3. Value Evaluation of Soil Conservation

Forests can reduce erosive rainfall and intercept, disperse, detain, and filter surface runoff with their unique canopy structure, large root tissue, and litter layer. At the same time, the forest enhances the content of soil humus and water-stable aggregates to stabilize the soil, reduce the loss of soil nutrients, and improve soil physical and chemical properties [33,34]. Soil conservation value evaluation can usually be analyzed through forest reduction of land abandonment, forest mitigation of sediment accumulation, and forest conservation of soil fertility [35].
(1)*Liquidambar formosana* plantations reduce the economic value of abandoned land. In this study, the amount of land abandonment reduction is calculated based on the difference between the amount of bare land erosion and the actual amount of loss of the *Liquidambar formosana* plantation. This study adopts the research results of Kang et al. (2001) [30]. The average soil erosion modulus of local broadleaf forest was 0.73 t/(hm^2^·a), which is used to replace the soil erosion modulus of the *Liquidambar formosana* plantation. The soil erosion modulus of bare soil is 37.58 t/(hm^2^·a). Therefore, the amount of soil erosion reduced by the *Liquidambar formosana* plantation ecosystem is about 36.85 t/(hm^2^·a). According to the survey, it takes half an hour to excavate and transport the unit earthwork a short distance, and the labor cost per hour is calculated as RMB 30, so the cost of excavating and transporting the unit volume earthwork is 15 RMB/m^3^. The average soil bulk density was 1.28 g/cm^3^. According to Equation (3), the economic value of land abandonment reduction by the *Liquidambar formosana* plantation is 431.67 RMB/(hm^2^·a).(2)Economic value of soil fertility preservation in the *Liquidambar formosana* plantation. Table 7 lists the contents of main soil fertility indicators of *Liquidambar formosana* plantation. The average contents of organic matter, total nitrogen, phosphorus, and potassium in the *Liquidambar formosana* plantation reached 2.93%, 0.11%, 0.02%, and 0.41%, respectively. According to the inventory data of local secondary resources [36], the average organic matter content of local forest land is 2.905%, the total nitrogen is 0.159%, the total phosphorus is 0.029%, and the total potassium is 0.346%. In contrast, the fertility of the *Liquidambar formosana* plantation in Changsha is at the average level of the local forest.

Diammonium phosphate contains 14% nitrogen and 15% phosphorus, and potassium chloride contains 50% potassium. According to the market survey, the local organic fertilizer price is 350 RMB/t, the market price of diammonium phosphate is 2500 RMB/t, and potassium chloride is 2600 RMB/t. According to Equation (4), the economic value of soil fertility maintenance of the *Liquidambar formosana* plantation is 1993.16 RMB/(hm^2^·a).
(3)Economic value of *Liquidambar formosana* plantation in mitigating sediment accumulation. The amount of silt reduction by forests can be calculated from the amount of soil lost. According to the study of Xiao et al. (2000) [26], about 24% of the soil lost by forests enters river channels. Therefore, the mitigation of sediment accumulation by the *Liquidambar formosana* plantation is about 8.83 t/(hm^2^·a). According to Equation (5), the economic value of mitigating sediment accumulation of the *Liquidambar formosana* plantation is 39.5 RMB/(hm^2^·a).

In conclusion, the economic value of soil conservation of the *Liquidambar formosana* plantation in Changsha is 2464.33 RMB/(hm^2^·a).

### 3.4. Value Evaluation of Air Purification

In this study, the air purification efficiency of the *Liquidambar formosana* plantation ecosystem in Changsha is evaluated from two aspects: sulfur dioxide absorption and dust catching. The sulfur absorption intensity and sulfur purification capacity of the plantation are 0.37% and 80.2 kg/(hm^2^·a), respectively. Since the dust catching effect of the forest is affected by various changing factors, especially with obvious seasonal changes, the dust catching ability of broad-leaved forest in China is estimated as 10.11 t/(hm^2^·a) [37]. Assuming that the operation cost of sulfur dioxide reduction is 100 RMB/(t·a), the investment and treatment cost of sulfur dioxide is 600 RMB/t, and the dust catching cost is 170 RMB/t [38]. According to Equations (6) and (7), the economic value of air purification of *Liquidambar formosana* plantation in Changsha is 1776.5 RMB/(hm^2^·a).

### 3.5. Value Evaluation of Carbon Fixation and Oxygen Release

The organic matter produced by the forest is partly stored in the trunks, branches, leaves, roots, and other organs. Some of them die and fall, and some are eaten by animals or returned to the soil through leaching and secretion [39]. The proportion of organic matter lost due to animal feeding, leaching, and secretion is small. Therefore, the net primary forest productivity in this study is calculated by biomass increment and litter biomass. Biomass involves trunk, branch, and root. According to relevant literature [36,40], the dry weight of 1 m^3^ wood in the tree layer of local broad-leaved forest is 0.45 t, and the ratio of branch to dry weight of broad-leaved forest is 18.3% and that of root is 21.4%. It can be concluded that the annual increments of tree layer biomass and litter biomass of broad-leaved forest are 0.484 t and 0.242 t (dry weight), respectively. Thus, the annual net primary productivity of the *Liquidambar formosana* plantation is 12.1 t/hm^2^. According to the local broad-leafed forest soil carbon density of 28.99 t/hm^2^ [41], the dry matter energy of plants in the *Liquidambar formosana* plantation is 19.67 t/(hm^2^·a), the O_2_ release is 14.5 t/(hm^2^·a), and the soil carbon fixation is 29 t/(hm^2^·a), in accordance with the photosynthesis formula.

The value of carbon fixation is evaluated by adopting the average value of China’s afforestation cost of 273.3 RMB/t C and the international carbon tax standard of 646.7 RMB/t C [42], and the price of oxygen release is calculated according to the average price of oxygen of 1000 RMB/t published by the Ministry of Health [43]. According to Equations (8) and (9), the economic value of carbon fixation and oxygen release is 36,703.33 RMB/(hm^2^·a).

### 3.6. Value Evaluation of Soil Improvement

The results of annual litter nutrient return in the *Liquidambar formosana* plantation are listed in Table 8. As depicted in the table, the annual return of main nutrients in the litter of *Liquidambar formosana* plantation is 59.215 kg·hm^−2^. Among the litter components, N concentration is the highest, and P content is the lowest. Diammonium phosphate contains 14% nitrogen and 15% phosphorus, while potassium chloride contains 50% potassium. According to the market price, diammonium phosphate is 2500 RMB/t, and potassium chloride is 2600 RMB/t. According to Equation (10), the economic value of soil improved by the *Liquidambar formosana* plantation is 930.83 RMB/(hm^2^·a).

### 3.7. Value Evaluation of Biodiversity Maintenance

A considerable number of species and quantity survey statistics are required to determine the conservation value of biodiversity by Shannon–Wiener index classification. Existing research results [44] are applied in this study. The Shannon–Wiener index grade and corresponding value quantity are listed in Table 9. In the study of Wang et al. (2008) [44], the Shannon–Wiener diversity index level of soft and broad-forest in Hunan Province is Grade II. Considering the particularity of biodiversity conservation in urban forest ecosystem, the conservation value of species diversity per unit area is corrected by 12,158.9 RMB/(hm^2^·a), and the average value is 26,079.50 RMB/(hm^2^·a).

### 3.8. Comprehensive Value and Analysis of Ecological Function

Based on the above analysis and value evaluation, the values of various ecosystem services of Changsha’s *Liquidambar formosana* plantation ecosystem are listed in Table 10. The total annual ecological value of the seven ecosystem services of the *Liquidambar formosana* plantation is 103,277.82 RMB/(hm^2^·a). The direct economic value is only 10,908.33 RMB/(hm^2^·a), accounting for 10.56% of the total value, while the indirect economic value is 92,369.49 RMB/(hm^2^·a), accounting for 89.44% of the total economic value. Furthermore, the indirect economic value is 8.47 times that of the direct economic value. This proportion structure is consistent with the results of many similar studies [24,28,42,43,45,46]. The comparison between the total ecological function value of the *Liquidambar formosana* plantation and other forests is shown in Table 11. The results show that the *Liquidambar formosana* plantation has high ecological function service value, which can provide reference for local ecological construction and development.

## 4. Discussion

Based on the above results, the total value of the seven ecosystem services provided by the *Liquidambar formosana* plantation is 103,277.82 RMB/(hm^2^·a), and the indirect economic value is significantly larger than that of the direct economic value. It indicates that the *Liquidambar formosana* plantation not only provides forest products and biological resources for human beings but also has ecological service functions in water conservation, environment purification, soil conservation, soil improvement, and biodiversity maintenance. Nearly 90% of its total economic value comes from the ecological protection value of the forest itself, which is the main part of the ecological function service value of the plantation. The relationship of the proportion reflects the huge ecological status of the plantation, which is conducive to the formulation of rational management plans and policies. As depicted in Figure 4, the seven ecological function service values are in the following order: carbon fixation and oxygen release > biodiversity maintenance> water conservation > organic matter production > soil conservation > air purification > soil improvement. The value of the carbon fixation and oxygen release function is the largest, accounting for 35.54% of the total value. Urban forest plants absorb carbon dioxide through photosynthesis and immobilize it in vegetation and soil [3]. At the same time, forest plants release oxygen to maintain the balance of carbon and oxygen in the air, which is of great significance for reducing the urban greenhouse effect [2,5]. The higher economic value of carbon fixation and oxygen release function produced by the *Liquidambar formosana* plantation not only reflects its strong photosynthesis but also reflects its significant carbon sequestration. The proportions of the economic value of biodiversity maintenance and water conservation are similar, with 25.25% and 23.64%, respectively. The *Liquidambar formosana* plantation is tall, with luxuriant branches and leaves. It not only provides a good living environment for various animals and understory vegetation but also becomes a natural shelter for a variety of creatures. Furthermore, relying on its wide branches and leaves, a large amount of litter and understory soil intercept and hold rainwater, which not only protects animal and plant resources but also plays a role in water conservation. *Forest plants* use solar energy to synthesize inorganic compounds into organic materials through photosynthesis, providing products for humans and other living things [5]. For the *Liquidambar formosana* plantation, the value of organic matter production is mainly reflected in the economic value of wood production. According to the evaluation, it can be seen that the *Liquidambar formosana* plantation grows rapidly with an average annual net accumulation of 12.89 m^3^/hm^2^, which reflects its strong productivity and is an excellent tree species in urban forests. The *Liquidambar formosana* plantation has great water conservation function, relying on the forest canopy, ground litter layer, and soil formed by its wide branches and leaves. Annual water conservation is 4272.83 t/hm^2^ and canopy interception > soil water storage > litter water capacity. Its regulation and storage function reduces flood disasters, increases the runoff in the dry season, and keeps the river water even and stable, prolonging the wet season. It also plays an active role in purifying water quality and increasing urban air humidity. Soil conservation, however, accounts for only 2.39% of the total economic value, but to keep the soil fertility, it also has an economic value of 1993.16 RMB/(hm^2^·a), an average equal to the local forest. In addition, soil is the foundation of forest growth. Without soil, forest ecosystems have no roots, so it is still important for soil to maintain its ecological function. In the evaluation of the economic value of air purification by the *Liquidambar formosana* plantation, due to the limitations of technology and equipment, the purification of harmful gases (e.g., nitrogen oxide, carbon monoxide, and chloride) by the forest is not considered, and the value of negative oxygen ion provided by the forest is not measured. Therefore, the proportion of economic value of forest air purification appears small. Compared with the other six ecological service values, the economic value of forest soil improvement accounts for only 0.90% of the total value. However, based on the annual return of main nutrients (59.215 kg·hm^−2^) of litter, the effect was still significant in improving soil functions, such as improving soil physical and chemical properties and providing a good biological environment. In the process of nutrient return, the return amounts are in the order N > K > P. The proportion of nutrients returned to leaves was the largest, accounting for 67.2% of the total. It is worth mentioning that this evaluation only analyzed and evaluated seven ecological service functions such as water conservation, soil conservation and air purification, taking into account the specific situation and operation feasibility of the *Liquidambar formosana* plantation in Changsha. However, ecosystem services such as forest recreation, windbreak and sand fixation, and microclimate improvement were not included in the evaluation system. In fact, the value of these ecological functions still exists, so the evaluation results in this paper may be slightly less than the actual economic value.

## 5. Conclusions

The total economic value provided by the *Liquidambar formosana* plantation was 103,277.82 RMB/(hm^2^·a). The ecological service values of the *Liquidambar formosana* plantation were sequentially: carbon fixation and oxygen release > biodiversity maintenance> water conservation > organic matter production > soil conservation > air purification > soil improvement. The function of carbon fixation and oxygen release accounted for 35.54% of the total value, which indicated the economic value generated by carbon fixation and oxygen release of the *Liquidambar formosana* plantation as well as its strong photosynthesis. Its carbon fixation function is of great significance. In addition, the indirect economic value was 8.47 times that of the direct economic value, significantly larger than that of the direct economic value. This proportional relationship reflects the huge ecological status of the ecosystem of the *Liquidambar formosana* plantation, which is conducive to formulating reasonable management plans and policies. 

## Figures and Tables

**Figure 1 ijerph-19-15317-f001:**
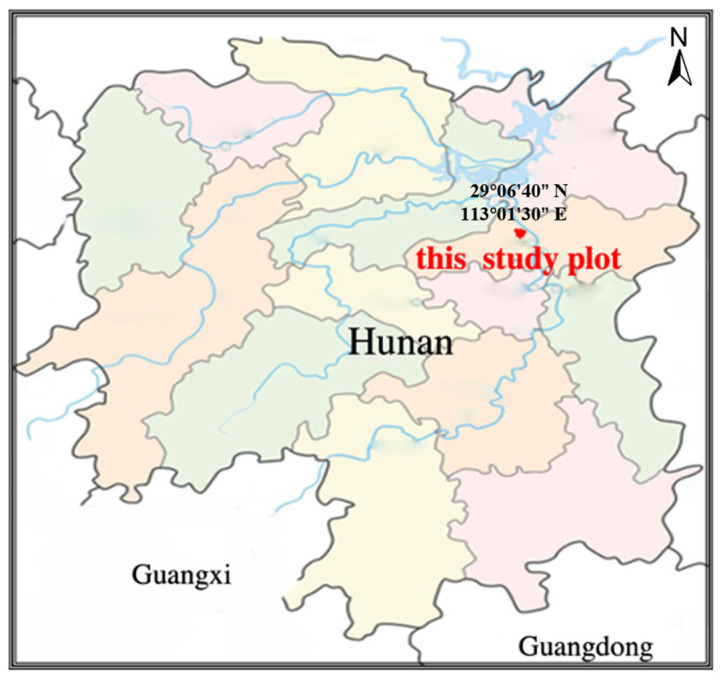
Location map of this study plot.

**Figure 2 ijerph-19-15317-f002:**
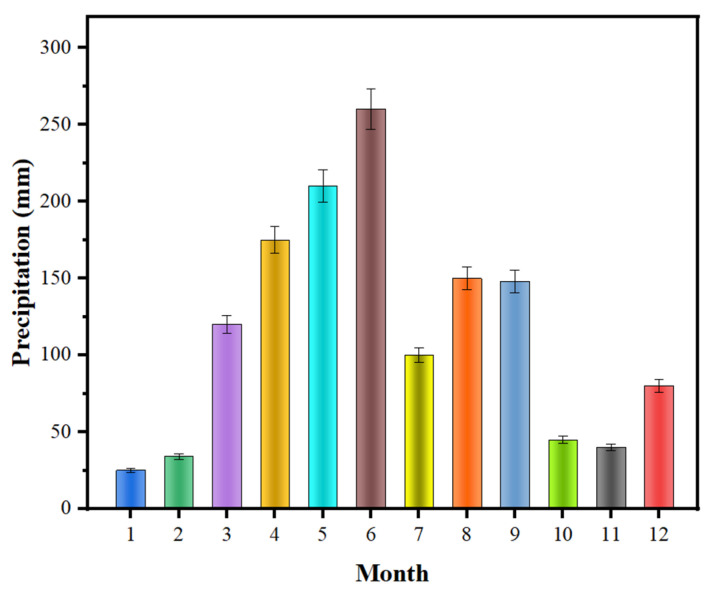
Monthly distribution of precipitation outside the forest (*Data from Changsha Meteorological Bureau*).

**Figure 3 ijerph-19-15317-f003:**
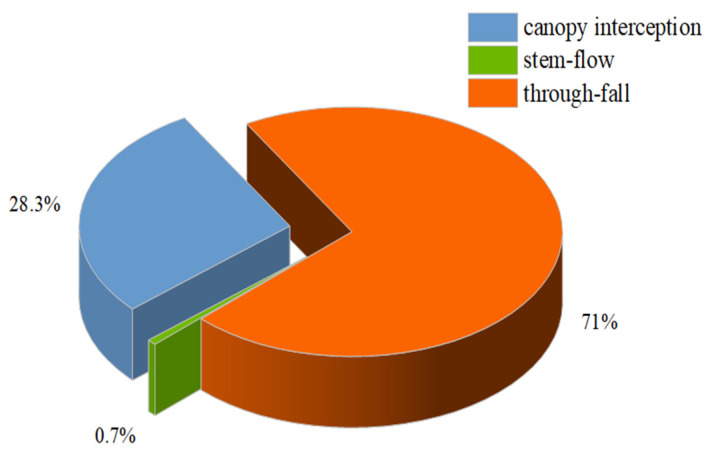
Canopy precipitation redistribution in *Liquidambar formosana* plantation.

**Figure 4 ijerph-19-15317-f004:**
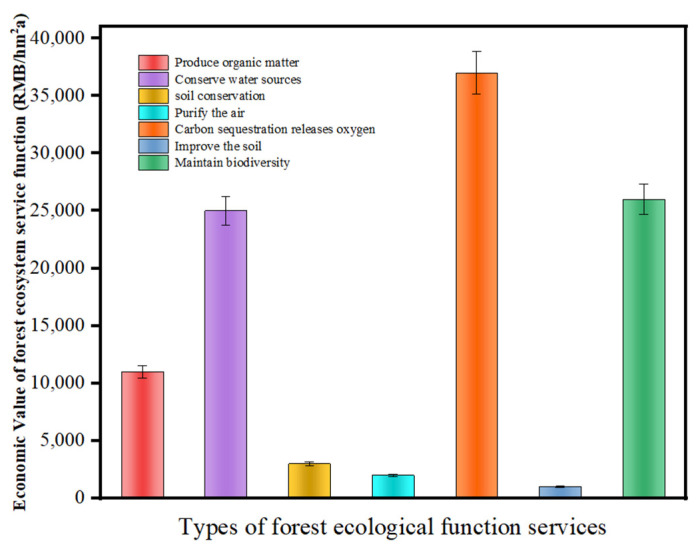
Ecological service function value of *Liquidambar formosana* plantation ecosystem.

**Table 1 ijerph-19-15317-t001:** Basic information of study plot.

Plot	Age of Stand (a)	Canopy Density	Density (N/hm^2^)	Height (m)	Diameter at Breast Height (cm)
I	24	0.80	1500	9.9	10.2
II	24	0.80	1475	11	10.01

**Table 2 ijerph-19-15317-t002:** Evaluation indexes and methods of main ecosystem services of *Liquidambar formosana* plantation.

Ecological Service Function	Evaluation Index	Method of Mass Calculation	Value Evaluation Method
Organic matter production	Wood production	Growth of living trees	Market value evaluation method
Water conservation	Annual regulated water volume	Forest canopy, litter, soil water capacity classification statistics	Shadow engineering method
Soil conservation	Soil fixation	Difference of soil erosion between forested land and bare land	Shadow engineering method
	Sediment accumulation mitigation	Calculation of soil retention and sediment deposition ratio	Shadow engineering method
	Soil fertility maintenance	Soil retention is determined by the nutrient content in the soil	Market value evaluation method
Air purification	Absorption of sulfur dioxide	Vegetation absorption capacity	Shadow engineering method
	Dust prevention	Vegetation absorption capacity	Shadow engineering method
Carbon fixation and oxygen release	Fixed carbon dioxide	Photosynthesis, soil absorption capacity	Carbon tax method
	Release oxygen	Photosynthesis	Market value evaluation method
Soil improvement	Litter nutrient return	Amount of nutrients in the litter	Market value evaluation method
Biodiversity maintenance	Forest conservation	Calculated according to Shannon–Wiener index	Market value evaluation method

**Table 3 ijerph-19-15317-t003:** Stocking volume of *Liquidambar formosana* Plantation in Changsha.

Plot	Age of Stand (a)	Stocking Volume of 2019 m^3^/(hm^2^·a)	Stocking Volume of 2020 m^3^/(hm^2^·a)	Stocking Volume of 2021 m^3^/(hm^2^·a)	Average Stocking Volume m^3^/(hm^2^·a)
I	24	40.18	53.68	66.90	12.89
II	24	47.38	60.20	72.23	

**Table 4 ijerph-19-15317-t004:** Litter layer storage [21].

Plot	Un-Decomposed Layer/kg·hm^−2^	Semi-Decomposed Layer/kg·hm^−2^	Total/kg·hm^−2^
*Liquidambar formosana* Plantation	2564.607	1463.608	4028.215

**Table 5 ijerph-19-15317-t005:** Litter water holding capacity.

Plot	Age of Stand (a)	Litter Water Holding Capacity (t/hm^2^)	Average Water Holding Capacity (t/hm^2^)
I	24	11.35	10.59
II	24	9.83	

The values of litter water holding capacity are calculated by multiplying the water absorption capacity of litter per unit mass by the total dry weight.

**Table 6 ijerph-19-15317-t006:** Forest soil water holding capacity of *Liquidambar formosana* plantation.

Plot	Soil Thickness	Natural Moisture Content (%)	Capillary Water Holding Capacity (mm)	Effective Water Storage (t/hm^2^)	Average Effective Water Storage (t/hm^2^)
I	0~15	17.96%	52.53	131.7	319.18
	15~30	17.79%	54.59	96.15	
	30~45	19.14%	53.31	90.0	
II	0~15	18.62%	53.39	132.0	
	15~30	20.24%	60.02	97.3	
	30~45	19.86%	57.42	91.2	

**Table 7 ijerph-19-15317-t007:** Main soil fertility indexes of *Liquidambar formosana* plantation [25].

Soil Layer Thickness (cm)	Capacity (g/cm^3^)	Organic Matter (g/kg)	Total Nitrogen (g/kg)	Total Phosphorus (g/kg)	Total Potassium (g/kg)
0~15	1.22	31.63	1.51	0.24	3.43
15~30	1.30	28.80	0.83	0.20	3.70
30~45	1.32	27.52	0.80	0.14	5.14
Average value	1.28	29.32	1.05	0.19	4.09

**Table 8 ijerph-19-15317-t008:** Annual return of litter nutrients in *Liquidambar formosana* plantations.

Component	N/kg·hm^−2^	P/kg·hm^−2^	K/kg·hm^−2^	Total/kg·hm^−2^	Proportion%
Leaf	31.587	1.140	7.090	39.817	67.2
Branch	9.781	0.345	1.660	11.786	19.9
Fruit	3.429	0.102	0.770	4.301	7.3
Debris	2.697	0.173	0.441	3.311	5.6
Total	47.494	1.760	9.961	59.215	100.0

**Table 9 ijerph-19-15317-t009:** Shannon–Wiener index grade and corresponding value.

Grade	Shannon–Wiener Index	Unit Price (RMB/hm^2^·a)
I	index ≥ 6	50,000
II	5 ≤ index < 6	40,000
III	4 ≤ index < 5	30,000
IV	3 ≤ index <4	20,000
V	2 ≤ index <3	10,000
VI	1 ≤ index <2	5000
VII	index ≤ 1	3000

**Table 10 ijerph-19-15317-t010:** Value of various ecological functions of *Liquidambar formosana* plantation ecosystem.

Function Type	Direct Economic Value	Indirect Economic Value							Total
	Organic Matter Production	Water Conservation	Soil Conservation	Air Purification	Carbon Fixation and Oxygen Release	Soil Improvement	Biodiversity Maintenance	Subtotal	
Evaluation result/ (RMB/year)	10,908	24,415	2464	1777	36,703	931	26,080	92,369	103,278
Proportion/%	10.56	23.64	2.39	1.72	35.54	0.90	25.25	89.44	100

**Table 11 ijerph-19-15317-t011:** Comparison of the total ecological function value of *Liquidambar formosana* plantation and other plantations.

Forest Species	The Total Ecological Function Value RMB/(hm^2^·a)	References
*Pinus massoniana Lamb.*	32,699.1	[47]
*Robinia pseudoacacia L.*	523,350.97	[48]
*Pinus tabuliformis Carrière + Robinia pseudoacacia L.*	50,660.4	[49]
*Fraxinus chinensis*	55,717.76	[50]
*Populus davidiana Dode + Robinia pseudoacacia L.*	46,436.2	[49]
*Cyclobalanopsis glauca*	220,800	[51]
*Pinus tabuliformis Carr.*	45,609.9	[52]
*Acrocarpus fraxinifolius Wight et Arn.*	89,067	[53]
*Platycladus orientalis (L.) Franco*	35,970.9	[52]
*Liquidambar formosana*	103,277.82	This work

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
