# Peer review of "Evaluation of Ecological Service Function of Liquidambar formosana Plantations"

_ijerph, 2022, doi:10.3390/ijerph192215317_

Round 1

Reviewer 1 Report

Dear Editor:

Thank you for giving me the opportunity to revise the manuscript entitled “Evaluation of Ecological Service Function of Liquidambar formosana Plantation” by Jingdong Wu and his/her colleagues that was submitted to “IJERPH”. The manuscript provided appropriate information about the studied task, but there are several requirements that have to consider by the authors. In this regard, the following comments are requested to be addressed by the authors:

Comment 1: The English of the paper is readable; however, I would suggest the authors to have it checked preferably by a native English-speaking person to avoid any mistakes.

Comment 2: The necessity & novelty of the manuscript should be presented and stressed in the “Introduction” section.

Comment 3: Provide a literature of the methods that used in survey at “Introduction”. The use of a table to demonstrate the advantage-disadvantage of these methods can be useful. Towards the end, mention the superiority & repeat the novelty of your work.

Comment 4: Please add a subsection clearly articulating the main limitations, wider applicability of your methods, and findings in the “Discussion” section.

Comment 5: The authors should deepen the discussion.

Comment 6: I would suggest that the authors review and include the following studies to improve the manuscript.

1. Azarafza, M., & Ghazifard, A. (2016). Urban geology of Tabriz City: Environmental and geological constraints. Advances in Environmental Research (AER): An International Journal, 5(2), 95-108.

2. Petrosino, P., Angrisani, A. C., Barra, D., Donadio, C., Aiello, G., Allocca, V., ... & Calcaterra, D. (2021). Multiproxy approach to urban geology of the historical center of Naples, Italy. Quaternary International, 577, 147-165.

3. Afzali, S., Sarmad, F. T., Heidari, M., & Jalali, S. H. (2021). Application of Urban Geology in Construction Projects (Case Study: Urban Geology of Sarpol-e Zahab, Kermanshah Province, Iran). Indonesian Journal of Geography, 53(1), 44-53.

Best regards,

Author Response

Dear Reviewer,

Best regards.

Reviewer 2 Report

Dear editor

The manuscript "Evaluation of Ecological Service Function of Liquidambar formosana Plantation" was submitted to International journal of Environmental Sciences and Public Health. It's a nice research about ecological services in China. I have some concerns and my decision is minor revision, finally I recommend to authors that improve the quality of this nice paper as following.

1.      The research results should be stated more accurately in the abstract. This will be of great help to the readers in the future and has an effective role in attracting the readers.

2.      There are repeated words between the title and keywords, please revise them, such as: Liquidambar formosana plantation; ecological service function.

3.      In the introduction, please try to write about the benefits of Liquidambar formosana in the field of human health. It is very important for the authors to respect the relationship between the research and the journal.

4.      "Survey of the Study Plot" change to "Study area".

5.      I could not find any references in lines 107-117.

6.      The results section should be separated from the discussion and written independently.

7.      One of the significant concerns is that the authors should carefully develop a discussion section to talk about the significance, shortages or advantages of the methods you proposed, the reliability and meaning of your results (compared to other related studies) etc.

8.      What advice and recommendations do you have for other researchers in future? Please revised conclusion, also it is so long.

9.         Please be sure that all the references cited in the manuscript are also included in the reference list and vice versa with matching spellings and dates.

10.  Finally, I checked plagiarism detection of this research and the similarity is 17% and there some concerns, please checked attached file.

Regards 

Author Response

Dear Reviewer,

Best regards.

Reviewer 3 Report

Full Title: Evaluation of Ecological Service Function of Liquidambar formosana Plantation

General comments: This paper applied the market value method, shadow engineering method, carbon tax method and other environmental economics methods to estimate the value of seven indirect ecological service functions of the urban forest. The aim of this work is to provide insights to the management, protection, and sustainable utilization of urban forest. I have a few concerns about the manuscript before I can recommend it being published.

Detailed comments:

1)      Line 9-15. The causality is not logic. This part needs to be better organized for highlighting the novelty of the work.

2)      Line 39. “Forest ecosystem lays the basis for human survival and development”? please explain more.

3)      Line 47-50. “Due to people's insufficient understanding of the service function and importance of the ecosystem, forest resources have been unreasonably exploited and utilized, thus leading to the destruction of the forest ecological environment [7].”

I don’t think so. Needs better wording.

4)      Line 54-57. I am not sure if the citation format is ok for IJERPH.

5)      Line 67. “In 1965, Professor Erik Jorgensen of the University of Toronto, Canada, put forward the formal concept of urban forest…”  Please cite the reference as a scientific work.

6)      Figure 1 needs notes of latitude and longitude.

7)      Section 2.1.1. you failed to say how did you collect or where did you get the related data (e.g., climate data and soil data), or dataset that was used to do your calculation later.

8)      Line 152.  “X2 is the soil erosion modulus (t·hm-2·a-1 ).” Of what?

9)      Line 179. “F is the annual 179 carbon fixation of forest soil per unit area (t);” how did you determine F here?

10)   Line 230 “After data sorting”, how did you do the data sorting process?

11)   Figure 2. Where did you get this data? You need to detail the dataset.

12)   Line 236-238 “, the canopy interception effect of Liquidb ambar formosana plantation in Changsha is stronger, more conducive to reducing the erosion effect of rainwater on soil”, why?

13)   Line 243 “The water holding capacity of forest litter is closely related to its storage”, I don’t understand the causal relationship between them.

14)   Line 246 “The water holding capacity of the litter is obtained by measuring the water absorption capacity of the litter on the forest land surface (Table 5)”, Needs more details.

15)   Data sources, and the analyzing methods should be described more clearly. For example, how “Litter water holding capacity” in table 5 was obtained? And where did you get the data in table 7? No matter the data was collected through a field investigation or from someone else, it is supposed to be clearly stated.

16)   Line 288-300, the comparisons were made only with local forestland, A comparison with bare land may be useful to highlight the contributes of the plantations’ effect.

17)   Table 9 “VII” “index” should be “<1”?

18)   There are no thoughtful discussions in this paper.

Author Response

Dear Reviewer,

Best Regards.

Round 2

Reviewer 1 Report

accepted.

Author Response

Thank you very much for your time and effort on managing our manuscript.

Reviewer 2 Report

Dear Editor

I did not find any tremendous changes in this revised version, I recommend the authors to strengthen the discussion section in the next version. In my opinion, the current version is not capable of being published in the desired magazine.

Regards

Author Response

Best regards.

Reviewer 3 Report

Thanks for the great effort that the authors put in improving the paper. I have no further concerns. 

Author Response

(The authors gave the same response as above.)
